# Is Laparoscopic Pancreaticoduodenectomy Feasible for Pancreatic Ductal Adenocarcinoma?

**DOI:** 10.3390/cancers12113430

**Published:** 2020-11-18

**Authors:** Chang Moo Kang, Woo Jung Lee

**Affiliations:** 1Division of Hepatobiliary and Pancreatic Surgery, Department of Surgery, Yonsei University College of Medicine, Seoul 03772, Korea; wjlee@yuhs.ac; 2Pancreatobiliary Cancer Center, Yonsei Cancer Center, Severance Hospital, Seoul 03772, Korea

**Keywords:** laparoscopic, pancreaticoduodenectomy, pancreatic cancer

## Abstract

**Simple Summary:**

Pancreatic cancer is known to be one of the most lethal malignant diseases in gastrointestinal tract. Margin-negative pancreatectomy followed by postoperative adjuvant chemotherapy is essential treatment for long-term survival. Due to anatomical complexity and technical difficulty, laparoscopic pancreaticoduodenectomy is still controversial. However, with the advance of laparoscopic surgery, laparoscopic pancreatic resection of pancreatic head cancer has been carefully applied in well selected patients. The accumulating data are suggesting its technical feasibility, safety, and potential equivalent long-term oncologic outcome. In this review, the current status of laparoscopic pancreaticoduodenectomy for pancreatic head cancer is summarized. In addition, potential surgical indications and future perspectives of laparoscopic pancreaticoduodenectomy for pancreatic cancer are discussed for safe implementation in our clinical practice.

**Abstract:**

Margin-negative radical pancreatectomy is the essential condition to obtain long-term survival of patients with pancreatic cancer. With the investigation for early diagnosis, introduction of potent chemotherapeutic agents, application of neoadjuvnat chemotherapy, advancement of open and laparoscopic surgical techniques, mature perioperative management, and patients’ improved general conditions, survival of the resected pancreatic cancer is expected to be further improved. According to the literatures, laparoscopic pancreaticoduodenectomy (LPD) is also thought to be good alternative strategy in managing well-selected resectable pancreatic cancer. LPD with combined vascular resection is also feasible, but only expert surgeons should handle these challenging cases. LPD for pancreatic cancer should be determined based on surgeons’ proficiency to fulfil the goals of the patient’s safety and oncologic principles.

## 1. Introduction

Pancreatic cancer is one of the most lethal malignant tumors in the human gastrointestinal tract. Its overall survival is reported to be around 5%. Until now, margin-negative pancreatectomy and postoperative adjuvant chemotherapy is known as the standard treatment option for cure of the disease [1,2]. However, most pancreatic cancer patients are found in the advanced cancer stage. Only about 15% of the patients are eligible for resection and more than half of the patients usually develop local or systemic recurrence within 2 years after surgery. The 5-year survival of the patients who underwent radical pancreatectomy is known to be around 20%. Recent statistical perspectives estimated that pancreatic cancer will be one of the top 3 cancers killing humans in 2030 [3].

The pancreas is a difficult internal organ to be accessed by minimally invasive surgery. It is located in retroperitoneal space, and major vascular structures are near the pancreas. Therefore, laparoscopic exposure and dissection of the pancreas are difficult and even small breakage of tributary vessels will result in massive bleeding to obscure a clear surgical field. In addition, in the case of pancreatic head lesions, laparoscopic management of remnant pancreas and resected bile duct is still a great hurdle to overcome for safe surgical procedure [4]. However, with the advance of laparoscopic technique and experiences, laparoscopic distal pancreatectomy (LDP) is regarded as a safe and standard approach in well-selected left-sided pancreatic tumor [5], and even laparoscopic pancreaticoduodenectomy (LPD) is carefully thought to be an appropriate surgical option to treat periampullary lesions [6].

In the past, the Yonsei criteria was suggested as appropriate tumor conditions for oncologically safe laparoscopic radical distal pancreatectomy (LDP) [7]. According to our experience, the Yonsei criteria was found to be not only selection criteria for LDP, but also a clinically detectable parameter to predict long-term survival of left-sided pancreatic cancer [8]. Carefully expanding indications for LDP is quite acceptable as long as patients’ safety and oncologic principles are maintained [9]. In the absence of randomized trials, uncertainty regarding the oncologic efficacy of LDP still exists. However, accumulating experience shows that LDP is associated with comparable survival, R0 resection, and use of adjuvant chemotherapy when comparing to open distal pancreatectomy (ODP) [10,11].

On the other hand, the current status of LPD has a long way to go in terms of technical and oncological safety. Unlike LDP, many surgeons are still within their learning curve period. Nickel et al. [12] performed a meta-analysis of randomized controlled trials comparing LPD and open pancreaticoduodenectomy (OPD) [13,14,15]. They concluded that at the current level of evidence, LPD shows no advantage over OPD except lower estimated intraoperative blood loss. Even though three currently available randomized control trials (RCT) were included for meta-analysis, lack of blinding of the patients and personnel assessing the main outcomes, and the learning curve issue should be considered when interpreting the results. Especially, pancreatic head cancer requires not only for skillful laparoscopic techniques but also wisdom to select appropriate patients for safe and effective margin-negative LPD, because curative resection is known to be basis for long-term survival of pancreatic cancer. Due to aggressive tumor biology and anatomical intimacy between the pancreas and major vascular structures, LPD for pancreatic cancer should be performed by expert surgeons who have already overcome of their learning-curve period [16,17,18]. It should be the last stage of LPD application in periampullary cancers after acquisition of full technical maturation.

It might be too early to generalize the potential oncological role of LPD in managing pancreatic head cancer, however several emerging articles from expert surgeons are providing future insight that LPD can be safe and effective in well-selected pancreatic cancer patients [19,20,21]. In fact, recent NCCN guideline version 1. 2020 pancreatic adenocarcinoma recommended surgical treatment by laparotomy or minimally invasive surgery as treatment for resectable pancreatic cancer [22]. LPD is no longer just a debatable issue in treating pancreatic cancer, but already regarded as one of the recommendable options in clinical oncology of pancreatic cancer. However, the articles presented as the basis for such a guideline were thought to be limited; they were published a long time ago, and were not focused on pancreatic cancer [23,24]. More detailed concerns about indications and understanding the current status of LPD in treating pancreatic cancer will be a strong background to increase the justification of clinical practice of LPD for pancreatic cancer.

## 2. Rationale of Minimally Invasive Pancreaticoduodenectomy for Pancreatic Cancer

### 2.1. Technical Feasibility

#### 2.1.1. Surgical Extent

Lymph node (LN) involvement is analyzed to be a very important prognostic factor in pancreatic cancer. Indeed, more than half of the resected pancreatic cancer showed LN metastasis and poor survival outcomes [25]. Then, for potentially clearing metastatic LNs, can extended pancreaticoduodenectomy (PD), clearing LNs around celiac, superior mesenteric artery (SMA), paraaortic, and hepatoduodenal ligament, as well as nerve plexus dissection around celiac, hepatic artery, and SMA, improve oncologic outcome in patients with resectable pancreatic cancer (Table 1)? Due to frequent recurrence and poor long-term oncologic outcomes following curative resection of pancreatic cancer, extended dissection was once advocated in surgical management of pancreatic cancer [26,27,28]. However, these are all retrospective observational studies, and selection bias should be considered when interpretating surgical outcomes. To optimize the surgical extent of PD in treating pancreatic cancer, researchers performed several important prospective randomized control studies to determine the optimal extent of surgical resection in treating resectable pancreatic cancer (Table 2).

When looking at the number of retrieved LNs, extended PD shows a higher number of retrieved LNs than standard PD (Table 2). However, a recent meta-analysis [29] to review oncologic outcomes of five randomized controlled trials comparing extended and standard lymphadenectomy in patients with PD for pancreatic cancer demonstrated that PD with standard dissection was safe (shorter operation time, less transfusion, less overall postoperative complications, and similar R0) and showed similar long-term survival outcomes to that of extended dissection (HR = 1.01 [95% CI: 0.77–1.34], *p* = 0.923). It is known that intraoperative transfusion [35] and postoperative complications [36,37] have an adverse impact on survival of resected pancreatic cancer, suggesting potential benefit of standard dissection in treating pancreatic cancer. Although the number of retrieved lymph nodes is reported to be one of the important prognostic factors in treating pancreatic cancer [38,39], recently lymph node ratio, and number of positive lymph nodes, “NOT total nodes examined”, were associated with overall survival in resected pancreatic cancer [40].

In summary, standard PD is not inferior and has comparable oncologic outcomes in treating resectable pancreatic cancer. Therefore, PD with routine extended lymph node dissection is not recommended due to higher morbidity and comparable long-term survival. According to the current technical availability of LPD, at least the extent of standard dissection is thought to be well achieved by current laparoscopic surgical technique (Figure 1).

#### 2.1.2. Retroperitoneal Margin

Cancer cell involvement in the resection margin is known to be associated with early tumor recurrence and poor long-term oncologic outcomes [41]. The pancreatic neck and head wrap the superior mesenteric vein (SMV)/portal vein (PV), and the uncinate process of the pancreas is elongated from the pancreatic head and extends behind the SMV/PV to contact the right lateral aspect of the SMA. There are abundant lymphatics and neural tissues around this area. Surgeons need to consider not only the pancreatic neck margin, but also the retroperitoneal margin, the so called SMA lateral margin, when performing radical PD for pancreatic cancer, because pancreatic cancer cells can invade and infiltrate along the nerve tissue in the pancreas toward major arterial systems (the SMA, common hepatic artery, and celiac axis) [42,43,44]. Therefore, even in case of pancreatic cancer that is very separated from the SMA, pancreatic cancer cells can invade SMA through this nerve tissue. The oncologic significance of retroperitoneal margin clearance has been reported [45,46]. Butler et al. [47] recently performed systemic review on the oncologic role of periadventitial dissection of SMA in affecting margin status after PD for pancreatic cancer. According to this review, it was suggested that positive margin was associated with decreased survival and SMA margin involvement was the most often, which ranged 15–45% of resected pancreatic cancer.

Surgical dissection of nerve tissue (retroperitoneal margin) around SMA is not that simple. Not uncommonly, it is usually difficult to exactly differentiate from pancreatic uncinated process and SMA due to abundant lymphatic tissue, inflammatory changes associated with pancreatitis, and neural tissue around SMA. These tissues are all intermingled altogether. In addition, it is difficult to expose the right lateral aspect of SMA because this area is behind the SMV-SV confluence. Lastly, there are abundant lymphovascular structures and even small breakage of tributary vessels around SMA and SMV usually give rise massive bleeding to prevent from further safe dissection for margin-negative resection.

Therefore, considering oncologic significance of retroperitoneal margin in treating pancreatic cancer, surgical design and plan to obtain cancer-free retroperitoneal margin is very important. There are several surgical techniques to secure retroperitoneal margin during LPD (Table 3). Rho et al. [48] introduced the potential application of indocyanine green (ICG) to facilitate the securement of the SMA lateral margin in laparoscopic PD, which is based on the idea that the pancreas is a well perfused organ and near infra-red light can detect ICG location through fluorescent illumination. When ICG is accumulated in the pancreatic head, near-infrared light can show visual differentiation between the uncinated process and surrounding soft tissues along the SMA (Figure 2), helping surgeons to obtain a negative retroperitoneal margin. The impact of long-term oncologic outcome of ICG-based SMA lateral border dissection remains to be investigated further, however it is thought to be a useful technique when performing LPD in resectable pancreatic cancer. Nagakawa et al. [49] introduced the technique to expose the inferior pancreaticoduodenal artery and SMA lateral border by proximal-dorsal jejunal vein (PDPV) pre-isolation, which is thought to be one of the approaches for standard PD. Moreover, Morales et al. [50] and Zimmitti et al. [51] demonstrate the surgical method to obtain the clear SMA margin by applying the SMA artery first approach. Despite requiring advanced laparoscopic skills, these methods are thought to be useful for margin-negative resection in selected cases of relatively advanced pancreatic cancer. Kuroki et al. [52] also introduced the concept of the pancreas-hanging maneuver by Penrose drain in managing SMA margin during LDP, however, they did not describe the R0 resection rate among the patients with this technique, and no patients with pancreatic head cancer were involved.

#### 2.1.3. Combined Vascular Resection

Pancreatic cancer usually is found at an advanced stage of cancer. In spite of resectable pancreatic cancer, due to anatomic intimacy between the pancreas and surrounding major venous vascular structures, unexpected portal vein (PV) and superior mesenteric vein invasion (SMV) is often encountered during the operation. In the past, several articles [53,54] concluded that the oncologic outcome of PD with combined venous vascular resection (PD-VR) is not favorable, however some others showed that PD-VR did not increase the complication rate and postoperative mortality [55,56]. PD-VR is no longer considered an absolute contraindication in pancreatic head cancer. It is regarded as a safe and feasible option to improve the resection rate of pancreatic cancer. Especially, the oncologic role of combined venous vascular resection became highlighted as the concept of neoadjuvant chemotherapy was introduced in managing borderline or locally advanced pancreatic cancer. However, Peng et al. [57] recently evaluated the value of PD with combine vascular resection (PD-VR) for pancreatic head cancer. In this meta-analysis, a total of 12,031 patients (2186 patients with PD-VR, 22.2%) from 30 published articles were investigated. In comparison with the PD group, it was analyzed that the PD-VR group had a lower R0 resection rate and higher rates of complications such as biliary fistula, reoperation rate, delayed gastric emptying, cardiopulmonary abnormalities, hemorrhage, in-hospital mortality, and 30-day mortality. In addition, the blood loss, operation time, and total length of hospital stay were higher in the PD-VR group, concluding that PD-VR for pancreatic cancer should be carefully considered by selected pancreatic surgeons.

Until now, only a few case reports [58,59,60,61,62] and case series [19,63,64,65] have been reported on the technical feasibility and safety of LPD with combined venous vascular resection (LPD-VR, Table 4). Tangential resection with stapler or hand suture or patch reconstruction appears to be common among the patients with LPD-VR (Figure 3), which is also confirmed when comparing with OPD-VR [19]. As shown above, postoperative morbidity and mortality related to combined vascular resection are not ignorable, and segmental resection and end-to-end anastomosis or artificial graft, or renal vein graft must be a challenging procedure. Therefore, LPD-VR also needs to be performed in well-selected patients and by highly selected expert surgeons.

The long-term oncologic efficacy of LPD-VR is lacking, but Croome et al. [19] demonstrated no statistical significance between LPD-VR and OPD-VR in terms of overall survival, leaving some room to be further investigated. In addition, there are other modified techniques for reconstruction following LPD-VR, using falciform ligament [66], parietal peritoneum [67], and hepatic ligament teres [68], suggesting further potential applications of laparoscopic technique remain to be studied.

#### 2.1.4. Repeated Pancreatectomy for Pancreatic Cancer in Remnant Pancreas

Even after R0 radical resection, patients with PC often experience local recurrence. However, it has been reported that about 0.5–4.6% of pancreatic cancers develop in remnant pancreas [69,70]. Since 1995, re-resection of pancreatic cancer in remnant pancreas has been proposed for improving oncologic outcome [71]. The first case series reporting 30 patients with surgery for recurred pancreatic cancer also suggested that resection for recurrent pancreatic cancer can be performed safely, and questioned a potential subgroup who might actually benefit form re-resection of recurred pancreatic cancer [72].

As experience of resected remnant pancreatic cancer accumulated, Yamada et al. [73] published the results of resection of recurrent remnant pancreatic cancer in Japanese society of hepato-biliary-pancreatic surgery. Clinical data from 114 patients with remnant pancreatic cancer after initial pancreatectomy were analyzed. It was found that median survival of the resected remnant pancreatic cancer was superior to the non-resected group (26 and 14 months, respectively [hazard ratio: 0.56, *p* = 0.012]), showing that re-resection of remnant pancreatic cancer could offer a favorable outcome and chance for cure. Similarly, the recent literature [69] reviewing 49 reported patients with resected remnant pancreatic cancer following resection of the primary pancreatic cancer demonstrated that median disease-free survival was 44.4 months (12–143 months), and median survival time was 32 months after repeated pancreatectomy. In addition, another systemic review demonstrated that re-resection of isolated local recurrent pancreatic cancer showed the most favorable survival outcomes (median, 32 months) comparing with chemoradiation therapy (19 months), stereotactic body radiation therapy (16 months) [74], suggesting the role of aggressive surgical extirpation of recurred remnant pancreatic cancer for improving prognosis. Table 5 summarized recent review articles reporting oncologic benefit of repeated pancreatectomy for pancreatic cancer. It was found that completion total pancreatectomy is the most common procedure for treating recurred pancreatic cancer in remnant pancreas and nearly no postoperative mortality is reported. Reported median survival time (14–32 months), and 5-year overall survival (40.6%) following repeated pancreatectomy are much longer than unresectable pancreatic cancer in recent RCTs [75].

Then, is laparoscopic repeated pancreatectomy feasible and does it provide a potential oncologic role in treating recurrent pancreatic cancer in remnant pancreas? Until now, very few reports have been published showing the technical feasibility of laparoscopic completion of total pancreatectomy for remnant recurrent pancreatic cancer following LPD for primary pancreatic cancer. Recently, Kang et al. (accepted in Ann Hepatobiliary Pancreat Surg 2020, and in process) successfully demonstrated the technical feasibility of laparoscopic repeated pancreatectomy for recurrent pancreatic cancer following initial laparoscopic radical pancreatectomy. In addition, Nagakawa et al. [76] and Sunagawa et al. [77] also showed the technical feasibility of laparoscopic resection of remnant pancreatic cancer after LPD for other primary cancer. All these reports are suggesting the technical feasibility and potential role of a minimally invasive approach even in recurrent pancreatic cancer in remnant pancreas. Further study is mandatory.

## 3. Current Literature

Recently, three meta-analyses have been published, comparing the oncologic outcomes of LPD and OPD. Jiang et al. [81] systematically reviewed the articles comparing LPD and OPD for the treatment of pancreatic cancer. They found 8 studies involving 15,278 patients and performed meta-analysis. It is thought that this meta-analysis was the first to evaluate clinical efficacy of LPD for the treatment of pancreatic cancer with long-term survival outcomes. It was found that there was no significant difference in the 5-year overall survival (HR: 0.97, 95% CI: 0.82–1.15, *p* = 0.76). In addition, LPD resulted in a higher rate of R0 resection, more harvested lymph nodes, shorter hospital stays, and less estimated blood loss, concluding that LPD is not inferior to OPD with respect to long-term oncologic outcomes, as well as better short-term surgical outcomes in patients with pancreatic cancer.

Chen et al. [82] also evaluated 1507 patients from 6 comparative cohort studies, comparing LPD with OPD for pancreatic cancer. Similar short-term oncologic outcomes were identified, such as lymph nodes harvested, R0 rate, number of positive lymph nodes, adjuvant treatment, and time to adjuvant treatment. Interestingly, in spite of comparable 1-year and 2-year survival, the following 3-year (OR 1.50, *p* = 0.007), 4-year (OR 1.73, *p* = 0.04), and 5-year survivals (OR 2.11, *p* = 0.001) were significantly longer in LPD group.

Yin et al. [83] identified 6 studies including 9144 pancreatic cancer patients and evaluated short-term and long-term oncological outcomes. They noted that fewer postoperative complications (*p* = 0.005), better trend of performance in R0 resection (*p* = 0.07), retrieved number of lymph nodes (*p* = 0.07), and comparable long-term survival (*p* = 0.49) were associated with LPD, concluding that LPD can be a suitable alternative to OPD in selected PDAC patients with respect to both surgical and oncological outcomes.

In addition, there are several articles analyzing the national cancer data base (NCDB) to overview the safety and effectiveness of LPD in treating pancreatic cancer. Sharpe et al. (study period: 2010–2011) [84] showed that, among the 4421 patients, a very limited number of patients (384 patients, 9%) underwent LPD for pancreatic cancer, and about one third of cases (118 patients, 30%) were performed in high volume centers (5 institutions, 3.8%). They demonstrated that LPD was equivalent to OPD in length of stay, R0-resection, lymph node count, and readmission rate, however a higher 30-day mortality rate in LPD group was noted in lower volume centers (7.5% vs. 3.4%, *p* = 0.003), raising concerns about the safety of LPD in treating pancreatic cancer due to a surmountable learning curve for the procedure. Kantor et al. (study period: 2010–2013) [85] analyzed 8213 patients with pancreatic cancer to show short-term and long-term oncologic outcomes of LPD and OPD. They found that LPD (828 patients, 10%) provides comparable short-term oncologic and long-term overall survival outcomes with OPD (20.7 months vs. 20.9 months, *p* = 0.68). In addition, decreased incidence of prolonged length of hospital stay (OR = 0.79), decreased rate of readmission and decrease in incidence of delay to adjuvant chemotherapy for LPD (OR 0.71, *p* = 0.11) were noted in LPD, suggesting a trend towards accelerated recovery. Higher 30-day mortality was still noted in LPD performed by lower volume centers (<20 LPDs, 5.6% vs. 0%, *p* < 0.01), however no significant differences of 30-day and 90-day mortality in high volume centers (≥20 LPDs) between LPD and OPD (0% vs. 1.3%, *p* = 0.08, and 0.6% vs. 3.4% *p* = 0.08, respectively). Chapman et al. (study period: 2010–2013) [86] evaluated the safety of LPD in elderly patients (≥75 years old, 1768 patients). Although it was found that more than 60% of the patients with LPD was performed in low volume centers (<5 LPDs), no significant differences were found in R0-status, number of lymph nodes examined, lymph node, status, receipt of adjuvant chemotherapy, days to initiation of adjuvant chemotherapy, readmission rates, or 30-day mortality between two groups (*p* > 0.05). Ninety-day mortality was significantly lower in LP (7.2 vs. 12.2%, *p* = 0.049). It was observed that there was a trend towards improved OS (HR = 0.85, *p* > 0.05) in the LPD group compared to the OPD group after adjusting for patient and tumor-related characteristics. Lastly, Torphy et al. (study period: 2010–2015) [87] also compared short-term and oncologic outcome of minimally invasive PD (MIPD) with those of OPD across low and high-volume centers. Among the patients with pancreatic cancer who underwent PD, 3754 patients (17.1%) were found to be performed minimally invasively. It was found that patients with MIPD for pancreatic cancer were less likely to stay in the hospital (OR, 0.75; 95% CI, 0.68–0.82). Thirty-day mortality, 90-day mortality, unplanned readmissions, margins, lymph nodes harvested, and receipt of adjuvant chemotherapy were equivalent between two groups.

Table 6 shows the recent significant articles comparing LPD and OPD in treating pancreatic cancer. When looking at the individual data, they are all pointing out that LPD can provide equivalent or superior short-term oncologic outcomes with comparable long-term survival outcome, suggesting that LPD is technically feasible and even oncologically safe in treating pancreatic cancer. It is interesting to note that every reported article demonstrates a smaller amount of intraoperative blood loss comparing with OPD. This observation is very important to improve the long-term oncologic outcomes of resected pancreatic cancer, because a lower volume of intraoperative estimated blood loss can lead to a lower chance of unnecessary intraoperative blood transfusion. Potential deleterious effects of blood transfusion on oncologic outcomes have been explained by several hypothesized mechanisms, primarily via the induction of immunosuppression [88,89]. In several studies, intraoperative transfusion was found to be one of the independent prognostic factors in resected pancreatic cancer [90,91,92]. In addition, a recent meta-analysis [35] to evaluate potential relationship between perioperative blood transfusion and prognosis of pancreatic cancer surgery also demonstrated detrimental effect of blood transfusion on survival in univariate (68.4%, 13 out of 19 studies) and multivariate analysis (47.4%, 9 out of 19 studies) respectively, showing overall blood transfusion associated with pancreatic cancer surgery can be related to shorter overall survival (pooled odds ratio 2.43, 95% confidence interval 1.90–3.10). Considering poor long-term oncologic outcomes of pancreatic cancer, intraoperative transfusion must be an attractive issue for pancreatic surgeons, because it is thought to be a surgeon-controllable factor by reducing intraoperative blood loss. In the near future, appropriate transfusion guidelines including strict transfusion threshold and active investigation on alternative to allogenic blood transfusion are necessary in managing pancreatic cancer patients.

However, all these articles showing comparable oncologic outcomes between LPD and OPD are based on retrospective studies, where unavoidable selection bias should be always concerned. Therefore, the advantages of laparoscopic PD on short-term surgical outcomes and comparable long-term survival rates should be regarded with careful interpretation.

However, unfortunately there is no randomized control trial (RCT) comparing LPD and OPD for pancreatic cancer. All conclusions were based on retrospective observational studies. Therefore, the interpretation of above these articles should be careful because there are unavoidable limitations, such as small sample size, selection bias, learning-curve issue, and heterogeneity in reported data.

Which approach, laparoscopic vs. open, is better regarding surgical approach to pancreatic cancer? An RCT is still needed to actually elucidate the true oncologic value of LPD, and to suggest standard care in treating pancreatic cancer. However, an RCT regarding this issue is not that easy to conduct because most pancreatic cancers are found to be at an advanced stage at diagnosis, and the surgical technique of LPD is not fully matured in the present surgical society. In fact, three RCTs comparing LPD and OPD were reported [13,14,15], but their study population included not only pancreatic cancer but also other periampullary cancer, benign and low-grade malignant tumors of the pancreas. Overall, they demonstrated no statistical differences, except less blood loss in LPD, and shorter operative time in OPD.

However, when looking at the last study recently performed by Hilst et al. [15], nine surgeons who had experienced more than 20 cases of LPD or OPD were involved. This clinical trial was prematurely terminated due to safety concerns of LPD. This phenomenon is thought to be a part of reflection to previous NCDB-based studies showing LPDs done in low volume centers were associated with higher mortality rate [84,85]. Taking the tumor biology of pancreatic cancer and its clinical presentations into consideration, associated pancreatitis, cholangitis, and potential risk of vascular involvements usually make laparoscopic oncologic dissection much more difficult and even dangerous. Therefore, LPD for treating pancreatic cancer should be a more challenging issue. All recent studies agree with the fact that many cases are required to overcome the learning curve for safe LPDs [16,17]. Therefore, under reasonable inclusion criteria, it is highly demanded that expert surgeons with good experience in both LPD and OPD should collaborate for a well-designed RCT to answer this question (ClinicalTrials.gov Identifier: NCT03870698). For a while, the currently accumulated expert surgeons’ experiences might be regarded as the highest level of evidence that we can take in current clinical situation of pancreatic cancer.

## 4. Proposal Potential Indications

Previously, we suggested a model for determining the indications of minimally invasive radical distal pancreatectomy for left-sided pancreatic cancer [7]. A similar approach will be possible in LPD for pancreatic cancer. Technical feasibility, procedural safety (or surgical risk), and surgical extent for margin-negative resection (oncologic clearance) should be considered in defining potential indication of LPD. A surgeon’s capacity of technical feasibility should cover the surgical extent requiring for curative resection. Figure 4 shows the dynamic relationships between these factors.

As the surgical extent for curative resection increases, the following will happen: First, the potential surgical risk may increase. Second, advanced surgical technique is highly required to ensure surgical extent for obtaining margin-negative resection. However, at some point, an appropriate surgical procedure for curative resection cannot be maintained by the laparoscopic approach due to surgeons’ own technical issues and patients’ co-morbidities. Therefore, there must be an optimum surgical extent that can be obtained by individual surgeons’ own surgical techniques for margin-negative resection.

From that point of view, three types of surgical extent of LPD can be available. *Type 0* LPD is PD with standard dissection. It does not require any type of combined vascular resection. *Type I* LPD requires combined venous vascular resection. In this surgical extent, most cases need tangential (wedge) resection of PV, or SMV with primary repair (*Type Ia* LPD). *Type Ib* LPD requires more complicated procedures for margin-negative resection, such as tangential resection with patch repair, segmental resection with end-to-end anastomosis or reconstruction using artificial graft (venous resection 2). In selected cases, combined arterial resection (*Type II* LPD) will be required, but quite limited [93].

For example, (A) surgeons’ technique for LPD is not enough to achieve curative resection in resectable pancreatic cancer. This group of surgeons need to do open PD for curative resection of the pancreatic cancer. However, (B) surgeons’ technique for LPD is good enough for obtaining margin-negative radical resection. They may perform LPD in well-selected cases expecting no combined vascular resection (Type 0 LPD). Only a few surgeons can perform LPD even with combined venous vascular resection (tangential or segmental) or even combined arterial resection in well-selected patients (Type I and II LPD). Therefore, considering the present technical feasibility to maintain patient’s safety, and margin-negative curative pancreatectomy, tumor conditions that could be removed by standard PD without combined vascular resection (Type 0 LPD) will be the primary indication for LPD. Others (Type I and II) can be performed in selected cases by only expert surgeons.

When looking at the anatomic relationship between pancreatic cancer and major vascular structures at diagnostic stage, anatomically “*resectable*” pancreatic cancer includes potential candidates who can undergo LPD for curative intent in clinical practice [94]. However, unexpected involvement of adjacent venous systems, such as superior mesenteric vein (SMV), or portal vein (PV) because of severe pancreatitis or tumor invasion, can be encountered during LPD for resectable pancreatic cancers, and this situation might be necessary for combined vascular resection. Only expert surgeons are responsible for this advanced stage of the pancreatic cancer. These cases mostly will result in elective conversion to open for procedural safety and curative resection [95].

Therefore, in the beginning stage, it is thought that only a small proportion of resectable pancreatic cancer without contact with the SMV-PV complex could be a potential indication for LPD (Type 0 LPD) for generalizing concept of LPD for pancreatic cancer (Figure 5). However, in the near future, with the advance of a diagnostic strategy for early detection and surgical techniques, potential candidates for LPD are certain to increase

## 5. Conclusions

It is worthy to emphasize that margin-negative radical pancreatectomy is the essential condition to obtain long-term survival of patients with pancreatic cancer. With the investigation for early diagnosis, introduction of potent chemotherapeutic agents, application of neoadjuvant chemotherapy, advancement of open and laparoscopic surgical techniques, mature perioperative management, and improved patients’ general conditions, survival of resected pancreatic cancer is expected to be further improved. OPD will still be mainstream in treating resectable pancreatic cancer. Particularly, OPD with anatomy-driven extended dissection will play significant role in treating advanced pancreatic cancer following neoadjuvant chemotherapy, which harbors the potential risk of combined major venous or arterial resection.

On the other hand, the technical and oncological feasibility of LPD for pancreatic cancer is still controversial. Considering difficult clinical circumstances to perform RCT, the currently reported experiences are thought to be the highest level of evidence that we can consider in managing pancreatic cancer. According to the literature, it can be carefully concluded that LPD is thought to be a good alternative strategy in managing well-selected resectable pancreatic cancer. LPD with combined vascular resection is also feasible, but only expert surgeons should handle these challenging cases.

In spite of potential advantages of the laparoscopic approach, it provides surgeons with fundamental limitations during surgical procedures, such as 2-D operative view, fulcrum-effect, limited motion of the instruments, attenuated touch sensation, and enhancing tremor. Theoretically, the robotic surgical system was introduced to overcome these hurdles. However, it is also true that many laparoscopic surgeons had already overcome these limitations. Therefore, surgical approach should be determined according to the surgeon’s expertise, patients’ general condition, and tumor biology. As long as surgical principles for pancreatic cancer are kept in mind, open, laparoscopic or robotic approach to PD for pancreatic cancer are expected to provide meaningful short- and long-term oncologic outcomes of resected pancreatic cancer.

Finally, in application of minimally invasive pancreatectomy for pancreatic cancer, patients’ safety and principles of surgical oncology should be kept in mind [97]. In fact, these two principles should be considered not only in LPD but also in OPD. Whether PD is performed by laparoscopic or open approach, surgical approach should be determined by a surgeon’s own technical expertise to fulfil these two goals.

## Figures and Tables

**Figure 1 cancers-12-03430-f001:**
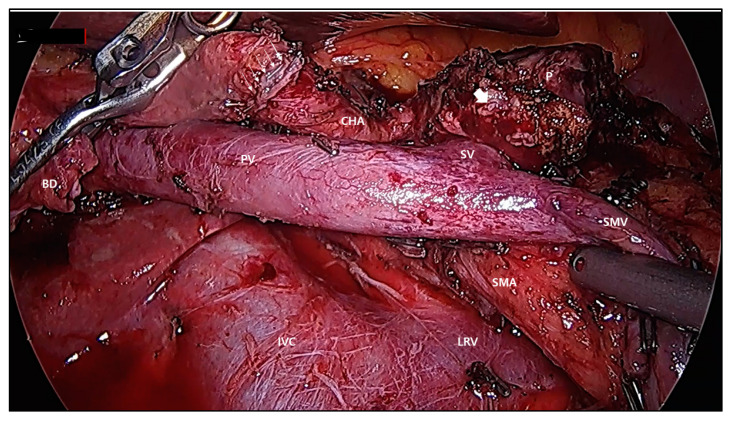
Laparoscopic view after resection of pancreatic head cancer. BD; bile duct, PV; portal vein, CHA; common hepatic artery. P; pancreas, SV; splenic vein, SMV; superior mesenteric vein, SMA; superior mesenteric artery, LRV; left renal vein, IVC; inferior vena cava, gastroduodenal artery stump (white arrows), pancreatic duct (thick white arrow).

**Figure 2 cancers-12-03430-f002:**
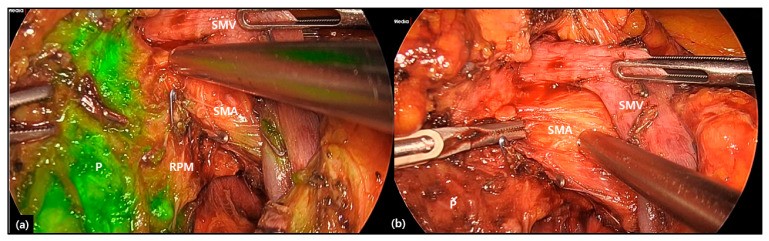
Intraoperative usage of indocyanine green (ICG) during obtaining the retroperitoneal margin of the pancreas. The ICG stained area is part of the uncinated process (**a**), which can be differentiated from the SMA to obtain the retroperitoneal margin (**b**). It can help the surgeon design surgical dissection. Note soft tissue around the SMA (retroperitoneal margin) is not stained by ICG. P; uncinated process of the pancreas, SMA; superior mesenteric artery, SMV; superior mesenteric vein, RPM; retroperitoneal margin.

**Figure 3 cancers-12-03430-f003:**
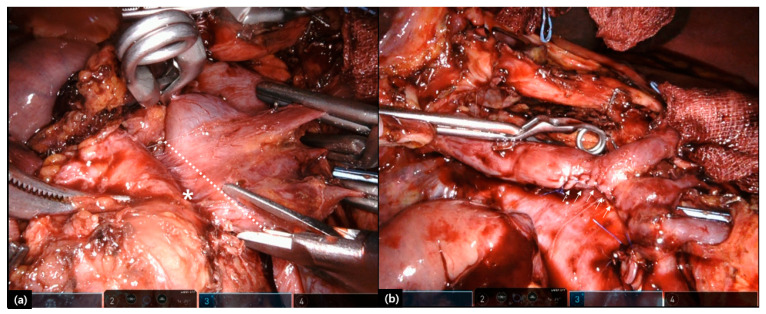
Combined venous vascular resection during laparoscopic pancreaticoduodenectomy (LPD). Tangential wedge resection of SMV is performed after transient clamping venous system (**a**). Primary repair of resected venous system (**b**). Tangential resection line (white dotted line), tumor invasion (white *), primary repair (white arrows).

**Figure 4 cancers-12-03430-f004:**
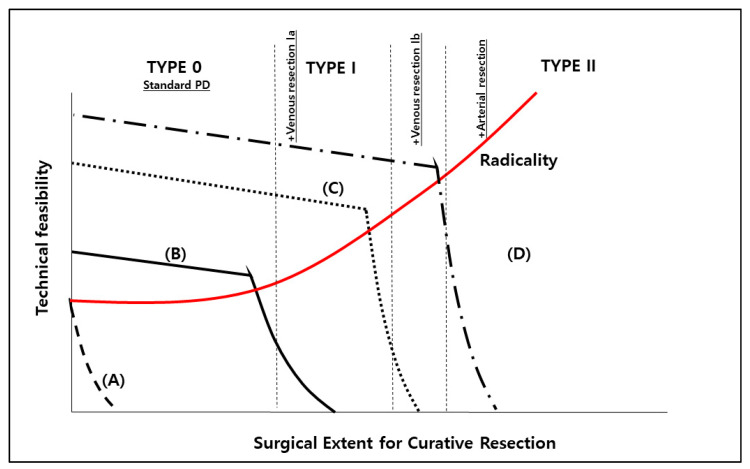
Model for determining surgical indication of LPD for pancreatic cancer. Regardless of laparoscopic or open PD for pancreatic cancer, surgical approach should be allowed only when surgeons’ technical feasibility can obtain the appropriate surgical extent for margin-negative resection. Therefore, indication of LPD can vary according to surgeons’ techniques and disease extent. OPD will be recommended in surgeon (**A**). Type 0 LPD can be done in surgeon (**B**). Type Ia LPD and Type Ib LPD can be allowed for surgeon (**C**) and surgeon (**D**), respectively. Anatomically *resectable* pancreatic cancer with intact fat plane between pancreas and major vascular structures is thought to be the ideal tumor conditions for LPD (Tumor conditions controlled by Type 0 LPD). Note: This author follows Fortner’s initial classification of regional pancreatectomy [96].

**Figure 5 cancers-12-03430-f005:**
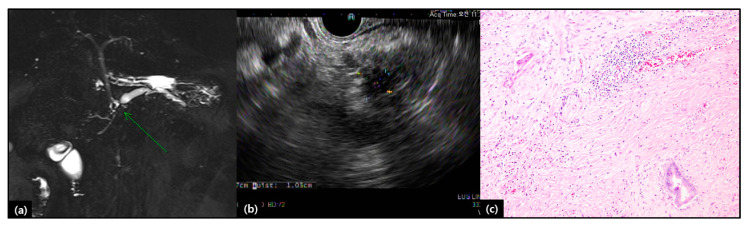
A case with extremely appropriate pancreatic cancer for LPD. A case of early pancreatic cancer in a 71-year-old, male patient, incidental finding of pancreatic duct dilatation during routine medical check-up. Preoperative image studies showed less than 1 cm sized pancreatic mass with secondary pancreatic duct dilatation (**a**,**b**). Type 0 LPD was performed on 24 October 2019, and discharged 9 days postoperatively. Pathological examination reported 3.0 mm-sized pancreatic ductal adenocarcinoma with good differentiation (×100, **c**). resected margins were free from malignant cells (safety margin ≥ 1 cm). No lymph node metastasis was noted among 23 retrieved lymph nodes.

**Table 1 cancers-12-03430-t001:** Topographic differences in extent of lymph node dissection between standard pancreaticoduodenectomy (PD) and extended PD.

Surgical Extent	Standard	Extended
LN station *	13, 17, 12b	13,17, 12b, 12a,12p, 8a, 8p, 5, 6, 9, 16, 14a, 14b,14c
Circumferential Nerve plexus dissection (Celiac/HA/SMA)	-/-/-	±/±/±

* Mentioned in more than 3 RCTs [29]; HA, hepatic artery; SMA, superior mesenteric artery.

**Table 2 cancers-12-03430-t002:** Literature review of randomized control trials (RCTs) comparing standard PD and extended PD in treating pancreatic cancer.

Author, Year	Surgical Extent	*N*	Op-Time, Hr/Min.	EBL, mL	#LNs	R0, %	LOH, Day	Cx., %	POPF, %	DGE, %	Mx., %	5YOS, %
Pedrazzoli, 1998 [30]	STD	40	371.9	2671	13.3	NA	22.7	47.5	12.5	NA	5	552d ^√^
	EXT	41	396.7	3149	19.8 *	NA	19.3	48.5	7.5	NA	4.9	589d ^√^
Yeo, 2005 [31]	STD	146(84)	5.9	740	17.0	88	11.3	29	6	6	4	10
	EXT	148(83)	6.4 *	800	28.5 *	93	14.3 *	43 *	13 *	16 *	2	25
Farnell, 2005 [32]	STD	40	6.2	NA	15	76	13	62.5	8	28	0	16
	EXT	39	7.6 *	NA ^∮^	34 *	82	16	34.1	13	36	3	17
Nimura, 2012 [33]	STD	51	426	NA	13.3	94.1	43.8	19.6	NA	NA	0	15.7
	EXT	50	547	NA	40.1 *	90	42.4	22.0	NA	NA	2	6.0
Jang, 2014 [34]	STD	83	355.5	372	17.3	85.5	19.7	32.0	9.6	9.6	0	18.8m ^√^
	EXT	86	419.6 *	563 *	33.7 *	90.7	22.8	43.0	12.8	5.8	2.3	16.5m ^√^

STD, standard dissection; EXT, extended dissection; EBL, estimated blood loss; LN, lymph node; LOH, length of hospital stay; POPF, postoperative pancreatic fistula; DGE, delayed gastric emptying; Mx, mortality; 5YOS, 5-year overall survival; *, *p* < 0.05; ^√^, median survival time; ^∮^, incidence of transfusion 5% vs. 23%, *p* = 0.01.

**Table 3 cancers-12-03430-t003:** Several approaches to obtain clear SMA lateral margin.

Author, Year	Concept	*N*	Op-Time	EBL, mL	R0, %	LOH, Day	Cx, %	Mx., %
Kuroki, 2010 [52] ^#^	Pancreas-hanging maneuver	9	NA	642	NA	19	NA	NA
Zimmitti, 2016 [51]	Periadventitial dissection of SMA	16	590	150	100	16	3	NA
Rho, 2018 [48]	ICG-guided differentiation of uncinate process from SMA lateral border	10	432	166	100	16.7	2	0
Nagakawa, 2018 [49]	PDJV preisolation method	21	489.38	183	NA	22.43	5	NA
Morales, 2019 [50]	Periadventitial dissection of for SMA TMpE	59	NA	NA	86	NA	NA *	NA

^#^ Pancreatic cancer was not included, * intraoperative massive bleeding in 3 cases, PDJV; proximal-dorsal jejunal vein, TMpE; total mesopancreas excision, SMA; superior mesenteric artery.

**Table 4 cancers-12-03430-t004:** Summary of articles reporting more than 5 cases of LPD-venous vascular resection (LPD-VR) for pancreatic cancer.

Author, Year	*N*	Op-Time, min	EBL, mL	Cx, %	Mx, %	LOH, Day	Survival
Cai, 2018 [63]	18(14) T: 8 E-E: 6 AG: 4	448	213	6(33.3)	None	13	Mean FU; 11 mo., 2 death (1 renal failure, 1 tumor metastasis)
Khatkov, 2017 [64]	8(5) T: 4 T+Patch:1 E-E: 3	560	450	2(25)	1(12.5) Due to heart failure	15	FU (4–12 mo.) 6 death due to tumor progression
Kendrick, 2011 [65]	11(9) T: 6 T+Patch: 4 RVG: 1	413	500	6	None	7	Median FU:7.2 mo. N/A
Croome, 2015 [19]	31(25) T: 12 T+Patch: 10 E-E: 7 RVG: 1 AG: 1	465	841.8	11(35)/2(6.4 *)	1(3.2)	6	Median FU : 15.2 mo. * no survival differences between OPD group (*p* = 0.14)

T, tangential resection; E-E, end-to-end anastomosis; RVG, renal vein graft; AG, artificial graft.

**Table 5 cancers-12-03430-t005:** Summary of review articles of repeated pancreatectomy for recurred pancreatic cancer in remnant pancreas.

Author, Year	N_Literatures_	N_Patients_	Initial Px (PP)PD/DP/ect)	Repeated Px TP/TP+PV/ect	Time Interval	Cx	Mx	Survival (From Initial Px/From Repeated Px)
Suzuki, 2019 [69]	17	49	31/17/1 (DPPHR)	44/4/1 (Segmentectomy)	12–143 month	NA	NA	114 months/ 32 months
Hashimoto, 2017 [78]	12	88	NA	78/5/5 (DP)	13–49 month	0–27%	0%	NA /14–27.5 months
Zhou, 2016 [79]	19	55	33/21/1 (DPPHR)	52/0/1 (DPPHR)	7–143 month	NA	0%	NA 5YOS, 40.6%
Choi, 2020 [80]	17	50	32/17/1	46/0/4 (partial)	18–88 month	NA	2%	107 months/ 60 months

Px, pancreatectomy; DP, distal pancreatectomy; NA, not available.

**Table 6 cancers-12-03430-t006:** Summarized recent articles investigating oncologic outcomes comparing between LPD and open pancreaticoduodenectomy (OPD) for pancreatic cancer.

Author, Years	Approach	N	Age	Gender (Male)	Op-Time	EBL	TF	Retrieved LNs	R0	Cx. (≥Clavien IIIB)	POPF	DGE	Mx.	LOH	Adjuvant CTx	Time to Adjuvant CTx	Survival	Anastomosis Technique
Croome, 2014 [19]	OPD	214	65.4	131	387.6	866.7	71	20.1	17	29	Grade B/C: 29(13.6)	Grade B/C: 26 (12)	4	9	164	59	25.3	
	LPD	108	66.6	51 *	379.4	492.4 *	21 *	21.4	84	6	6(5.6)	39(18)	1	6 *	82	48 *	21.8	Robot: 5, open: 7, lapa: 96, No detail description
Conard, 2017 [92]	OPD	25	66	18	NA	NA	10	17	21	III(A+B):11 IV:4 V:2	Yes: 10 No: 15	NA		27	11	NA	29.6	
	LPD	40	68	26	NA	NA	14	18	35	III(A+B):12 IV:10 V:2	Yes: 12 No: 28	NA		24.5	14	NA	35.5	ALL Laparoscopic reconstruction, E-to-S DTM internal stent PJ
Stauffer, 2017 [93]	OPD	193	68.9	96	375	600	90	17	154	58	Total:20 A:6 B:9 C:5	Total:28 A:12 B:6 C:10	10	9	122	55	20.3	
	LPD	58	69.9	32	518 *	250 *	20 *	27 *	49	13	Total:6 A:2 B:4 C:0	Total:10 A:4 B:3 C:3	2	6	41	54	18.5	Open conversion: 14, Lapa: 44, Two-layer DTM PJ
Kuesters, 2018 [94]	OPD	278	68	137	428	NA	65	16	195	107	NA	NA	6	16	NA	NA	18	
	LPD	62	71	31	477 *	NA	9	17	54 *	25	NA	NA	3	14 *	NA	NA	22	All Open reconstruction No detail description
Choi, 2020 [21]	OPD	34	63.3	18	471.2	448.8	2	20.6	24	NA	None:24 BL:6 B:3 C:1	None:30 A:2 B:2 C:0		19.9	27	55.1	44.6	
	LPD	27	63.3	12	477.7	232.5*	0	13.3	25	NA	None:17 BL:8 B:1 C:1	None:23 A:3 B:0 C:1		21.1	21	59.5	45.2	All Laparoscopic reconstruction, E-to-S DTM internal stent PJ
Zhou, 2019 [95]	OPD	93	64	68	260	200	7	11	88	13	Total:86 BL:74 B/C:12	Total:66 A:27 B/C:39	2	14	47	43.5	20	
	LPD	55	63	40	330*	150*	16 *	18 *	55	6	Total:20 BL:13 B/C:7	Total:1 A:0 B/C:1	0	13	26	39	18.7	All laparoscopic reconstruction, E-to-S DTM internal stent PJ

* *p* < 0.05, TF; transfusion; LNs; lymph nodes, POPF; postoperative pancreatic fistula, DGE; delayed gastric emptying, CD; Clavien-Dindo Classification, CTx; chemotherapy, DTM; duct-to-mucosa anastomosis, PJ; pancreaticojejunostomy.

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
