# Peer review of "Is Laparoscopic Pancreaticoduodenectomy Feasible for Pancreatic Ductal Adenocarcinoma?"

_cancers, 2020, doi:10.3390/cancers12113430_

Round 1

Reviewer 1 Report

This work reports the state of the art of laparoscopic pancreaticoduodenectomy. The subject is very topical. The presentation is clear, the writing of quality with an well documented. This work is newsworthy.

Two remarks that could enrich this work:

1 / the method of performing anastomoses is not described: by laparoscopic route or by laparotomy as part of a hybrid procedure? What type of pancreatic anastomosis was used in the studies analyzed?

2 / place of laparoscopic pancreaticoduodenectomy by robotics?

Author Response

We tried to respond to the reviewer 1's comments.

Hoping thie revised version to be improved.

BEST

CMK

Reviewer 2 Report

This is a detailed review comparing laparoscopic and open PD for pancreatic ductal adenocarcinoma. PD is a difficult procedure with a high complication rate even when performed by experienced surgeons. Laparoscopic PD is a more challenging operation and whether it may offer some advantages over the open technique is still unclear.

The Authors have correctly underlined that all conclusions in the literature were based on retrospective studies. Selection bias is likely to be occurred, therefore advantages of laparoscopic PD on short-term surgical outcomes and comparable long-term survival rates between the two techniques should be regarded with caution.

If possible, data on specific complications of pancreatic surgery (POPF and DGE) should be included in the evaluation of short-term outcomes.

In the analysis of data from the national cancer database I suggest including Torphy et al Ann Surg 2019.

Several abbreviations have been used without explanation. Median survival time and 5-year overall survival for repeated pancreatectomy (line 240) are not reported correctly. There are some typing errors and repeated words throughout the text.

Author Response

We tried to improve our manuscript based on reviewer 2's comments.

BEST

CMK
